# Can Blood Ammonia Level, Prehospital Time, and Return of Spontaneous Circulation Predict Neurological Outcomes of Out-of-Hospital Cardiac Arrest Patients? A Nationwide, Retrospective Cohort Study

**DOI:** 10.3390/jcm11092566

**Published:** 2022-05-04

**Authors:** Tsuyoshi Nojima, Hiromichi Naito, Takafumi Obara, Kohei Ageta, Hiromasa Yakushiji, Tetsuya Yumoto, Noritomo Fujisaki, Atsunori Nakao

**Affiliations:** 1Department of Emergency, Critical Care and Disaster Medicine, Okayama University Graduate School of Medicine, Dentistry and Pharmaceutical Sciences, Okayama 700-8558, Japan; t.nojima1002@gmail.com (T.N.); dainosinn@gmail.com (T.O.); ageage1982@gmail.com (K.A.); amorihamorih1111@yahoo.co.jp (H.Y.); tyumoto@cc.okayama-u.ac.jp (T.Y.); ntfujisaki@gmail.com (N.F.); qq-nakao@okayama-u.ac.jp (A.N.); 2Department of Primary Care and Medical Education, Okayama University Graduate School of Medicine, Dentistry and Pharmaceutical Sciences, Okayama 700-8558, Japan; 3Yakushiji Jikei Hospital, Okayama 719-1126, Japan

**Keywords:** ammonia, cardiopulmonary resuscitation, neurological outcome, biomarkers

## Abstract

Background: This study aimed to test if blood ammonia levels at hospital arrival, considering prehospital time and the patient’s condition (whether return of spontaneous circulation [ROSC] was achieved at hospital arrival), can predict neurological outcomes after out-of-hospital cardiac arrest (OHCA). Methods: This was a retrospective cohort study on data from a nationwide OHCA registry in Japan. Patients over 17 years old and whose blood ammonia levels had been recorded were included. The primary outcome was favorable neurological outcome at 30 days after OHCA. Blood ammonia levels, prehospital time, and the combination of the two were evaluated using the receiver operating characteristic curve to predict favorable outcomes. Then, cut-off blood ammonia values were determined based on whether ROSC was achieved at hospital arrival. Results: Blood ammonia levels alone were sufficient to predict favorable outcomes. The overall cut-off ammonia value for favorable outcomes was 138 μg/dL; values were different for patients with ROSC (96.5 μg/dL) and those without ROSC (156 μg/dL) at hospital arrival. Conclusions: Our results using patient data from a large OHCA registry showed that blood ammonia levels at hospital arrival can predict neurological outcomes, with different cut-off values for patients with or without ROSC at hospital arrival.

## 1. Introduction

Despite improvement in managing out-of-hospital cardiac arrest (OHCA), the proportion of patients with favorable neurological outcomes after suffering OHCA remains low [1]. Identifying neurologically intact survivors at an early stage is a high priority so clinicians can better inform families who need to make decisions about patient care and avoid early withdrawal of aggressive care because of “perceived” unfavorable neurological prognoses [2]. Current guidelines recommend that neurological prognostic tests are reviewed 72 h after the onset of cardiac arrest (CA), because no single test or clinical sign is satisfactory to predict neurological outcomes early after CA [2,3].

To determine neurological outcomes, serum biomarkers appear more objective and feasible compared with physical examinations such as pupillary reflex to light, corneal reflexes, and motor response or physiological tests including electroencephalogram (EEG) and evoked potentials [4,5,6,7,8]. Several studies have demonstrated that serum ammonia levels could be a useful biomarker for predicting the neurological outcomes of OHCA patients [9,10]. To the extent that liver function is preserved, an elevation of blood ammonia levels in CA patients is supposed to be attributed to suppression of the glycolytic pathway in red blood cells resulting from prolonged acidosis and hypoxia [11,12]. Theoretically, blood ammonia levels thus may be affected by prehospital time and differ significantly between patients with and those without return of spontaneous circulation (ROSC) at hospital arrival. We therefore hypothesized that blood ammonia level combined with these factors may be a better predictive indicator of favorable patient outcomes after OHCA.

Accordingly, the objectives of this study, which included 7426 OHCA cases from a nationwide registry in Japan, were to (1) evaluate whether blood ammonia levels at hospital arrival in conjugation with total prehospital time outperform blood ammonia levels alone in predicting favorable neurological outcomes after OHCA, and (2) determine cut-off blood ammonia levels to predict these outcomes, stratified by patients who either did or did not achieve ROSC at hospital arrival.

## 2. Materials and Methods

### 2.1. Study Design

This was a retrospective, observational, cohort study using data from the Japanese Association for Acute Medicine (JAAM) OHCA Registry. The Okayama University Ethics Committee approved the study (K2106-008) and waived the requirement for informed consent.

### 2.2. Data Collection

The JAAM OHCA Registry is a prospective, multicenter, web-based registry started in 2005. In 2021, it was expanded to 101 tertiary hospitals in Japan that provide critical and emergency care with the goal of better understanding the characteristics and treatment of OHCA patients [13]. All OHCA patients transported directly to participating facilities are registered in the database. Data collection and design of the JAAM OHCA registry have been previously described in detail [14]. Prehospital data, including time measurements (transportation date, transportation time, on-scene time, response time, total prehospital time, prehospital treatment time) for OHCA patients were collected by emergency medical services (EMS) personnel using radio-controlled watches and subsequently entered into the national registry of the Fire and Disaster Management Agency. Total prehospital time was defined as the time from EMS call to hospital arrival. Physicians at each hospital are responsible for using a form to collect and record data including each patient’s baseline characteristics (age, sex, cause of CA, etc.), information on prehospital setting (witnessed collapse, bystander cardiopulmonary resuscitation [CPR]), and treatments/outcomes at hospital (initial rhythm at hospital arrival, ROSC or sustained CA at hospital arrival, blood test results on hospital arrival, outcomes in hospital, etc.). The registry does not include any information about comorbidities such as hypertension, diabetes, cardiovascular disease, chronic kidney disease, and chronic liver disease. Blood ammonia levels were obtained at hospital arrival if applicable. Cerebral performance category (CPC) scale score at 30 days was used to determine neurological outcomes. Finally, data from the JAAM OHCA registry was integrated with data from the prehospital registry of the Fire and Disaster Management Agency and in-hospital data.

### 2.3. The Emergency Medical Services System in Japan

The EMS system in Japan has been described in detail elsewhere [15]. Briefly, all emergency calls (via 119 in Japan) are handled by local operations centers that dispatch the nearest ambulance to the scene. Each vehicle is staffed by three or four EMS personnel, at least one of whom is highly trained and known as an emergency life-saving technician (ELST). ELSTs can perform advanced airway management including supraglottic airway placement. In addition, specially trained ELSTs are allowed to perform endotracheal intubation and adrenaline administration under real-time medical direction by physicians. EMS personnel are obligated to resuscitate and transport OHCA patients to the hospital unless obvious signs of death are present.

### 2.4. Participants, Groups, and Endpoints

All patients with OHCA of cardiac and noncardiac causes treated from 1 July 2014 to 31 December 2017 in the JAAM OHCA registry were eligible for inclusion in the study. Patients over 17 years of age whose blood ammonia levels had been recorded at hospital arrival were included in the study. Patients with unknown ages were excluded.

Patients were divided into two groups based on CPC scores 30 days after OHCA; the favorable outcomes group and the poor outcomes group was comprised of patients with CPC scores of 1 or 2 and 3 through 5, respectively.

The primary outcome was favorable neurological outcome. The impact of blood ammonia levels on the primary outcome was determined using multiple logistic regression analysis. Then, prognostic performance for a favorable neurological outcome was determined using the receiver operating characteristic (ROC) curve. 

The relationship between blood ammonia levels and total prehospital time was examined for favorable or poor neurological outcomes. Then, the relationship between blood ammonia levels and total prehospital time was examined separately for the ROSC group and the non-ROSC group at the time of hospital arrival. For further analysis, patients were stratified into groups for OHCA with cardiac causes and OHCA with noncardiac causes.

### 2.5. Statistical Analysis

Continuous variables are described using medians with interquartile ranges. Categorical variables are described using numbers and percentages. We used multiple logistic regression analysis to identify independent predictors of favorable neurological outcomes. In addition to age, sex, total prehospital time, and blood ammonia levels on arrival, clinically relevant factors and covariates based on the results of univariate analysis were used to adjust for outcomes in multivariate logistic regression. The results of logistic regression are described using odds ratios (OR) with 95% confidence intervals (CI). Prognostic performance (area under the curve; sensitivity; specificity) was examined using total prehospital time and blood ammonia level at hospital arrival, respectively. The cut-off values for favorable outcomes were examined for blood ammonia levels and transportation time using Youden’s index. The association between time and blood ammonia level was determined with linear logistic regression and 95% CI. Stata version 16 (StataCorp LP, College Station, TX, USA) was used for the analysis.

## 3. Results

### 3.1. Patient Characteristics

A total of 34,754 OHCA patients were registered during the study period. After eligibility screening, 7426 patients were included. Favorable neurological outcomes were seen in 364 patients; poor neurological outcomes were seen in 7062 patients at 30 days after CA (Figure 1). 

Participants’ characteristics are shown in Table 1. The median age was 76 (64–84), and 4538 (61%) patients were male. Of the 7426 patients, 3181 (47%) had collapse witnessed by bystanders; bystander CPR was performed on 3202 (47%), and public automated external defibrillator (AED) was used at the scene on 149 (2.2%). Shockable rhythms, including pulseless ventricular tachycardia or ventricular fibrillation, were observed in 392 (5.3%) patients. The overall median total prehospital time was 34 (28–41) minutes, and median total prehospital times were 31 (25–39) minutes in the favorable outcomes group and 34 (28–42) minutes in the poor outcomes group. The overall median blood ammonia level at hospital arrival was 253 (125–438) μg/dL, and median blood ammonia levels were 70 (42–122) μg/dL in the favorable outcomes group and 265 (138–452) μg/dL in the poor outcomes group. ROSC on arrival was observed in 910 patients (12%). 

### 3.2. Impact of Blood Ammonia Level and Time on Favorable Neurological Outcomes

Multivariable logistic regression analysis for favorable neurological outcomes was performed and used blood ammonia levels and total prehospital time with other variables (age, sex, witnessed collapse, bystander CPR, AED use on scene, adrenaline use on scene, electrocardiogram on arrival, potassium and lactate levels at hospital arrival, and cardiogenic arrest) as adjustments. Lower blood ammonia levels indicated more favorable neurological outcomes (adjusted OR: 0.991, 95% CI: 0.989–0.993) and shorter total prehospital time indicated more favorable neurological outcomes (adjusted OR: 0.996, 95% CI: 0.997–0.997) (Table 2). 

### 3.3. Prognostic Performance of Blood Ammonia Levels

Figure 2 demonstrates the prognostic performance of blood ammonia levels. The areas under the ROC curve for total prehospital time and blood ammonia levels for the prediction of poor outcomes were 0.587 (95% CI: 0.551–0.622) and 0.849 (95% CI: 0.829–0.869), respectively. The cut-off values for each variable using Youden’s index were as follows: total prehospital time, 31.5 min (sensitivity 0.61, specificity 0.53, positive predictive value 0.96, negative predictive value 0.07), and blood ammonia levels, 138 μg/dL (sensitivity 0.75, specificity 0.80, positive predictive value 0.99, negative predictive value 0.16, positive likelihood ratio 3.74). The area under the ROC curve for blood ammonia levels at hospital arrival in combination with total prehospital time was 0.849 (95% CI: 0.829–0.870), which was similar to the ROC curve for blood ammonia levels alone.

### 3.4. Correlation between Blood Ammonia Level and Total Prehospital Time

Figure 3 depicts the correlation between the blood ammonia levels and the duration of total prehospital time in the favorable outcomes group and the poor outcomes group. The favorable outcomes group showed a slight trend towards increase in blood ammonia levels as prehospital time increased, while a stronger correlation was noted in the poor outcomes group; blood ammonia levels were markedly increased as a result of prolonged prehospital time. The blood ammonia levels of patients in the poor outcomes group were significantly higher compared to those in the favorable outcomes group, regardless of the duration of total prehospital time.

To further examine the impact of blood ammonia levels on patient outcomes based on the assumption that sustained CA could lead to hyperammonemia, the patients were divided into groups for those who had achieved ROSC and those who did not achieve ROSC at hospital arrival. Table 3 shows the basic demographics of the two groups. Blood ammonia levels were significantly lower in patients with ROSC compared to those without ROSC at the time of hospital arrival (103 vs. 278 μg/dL, *p* < 0.01). The patients who had achieved ROSC had better functional outcomes compared to those who had sustained CA (27 vs. 1.8%, *p* < 0.01). In patients who had obtained ROSC at hospital arrival, the blood ammonia levels of each outcome were almost constant regardless of total prehospital time (Figure 4A). Blood ammonia levels were significantly lower in patients with favorable outcomes compared to those with poor outcomes (70 vs. 265 μg/dL, *p* < 0.01). In patients who had sustained CA at the time of hospital arrival, blood ammonia levels were positively correlated with total prehospital time for both favorable and poor outcomes (Figure 4B). Similarly, for the group that had sustained CA, blood ammonia levels were significantly lower in patients with favorable outcomes than in those with poor outcomes (149 vs. 365 μg/dL, *p* < 0.01). The blood ammonia level cut-off points for favorable outcomes in the ROSC group and the sustained CA group were 96.5 μg/dL (sensitivity 0.65, specificity 0.76), and 156 μg/dL (sensitivity 0.56, specificity 0.64), respectively.

### 3.5. Cardiac Causes vs. Noncardiac Causes

Further analysis was performed to determine whether there were differences between patients with OHCA from cardiac vs. those with OHCA from noncardiac causes. Patients were divided into two groups: the cardiac causes group and the noncardiac causes group. Patients’ characteristics are presented in Appendix A. Multivariable logistic regression analysis revealed that lower blood ammonia levels were associated with favorable neurological outcomes in both the cardiac causes group and the noncardiac causes group (Appendix A). The areas under the ROC curve for the cardiac causes group and the noncardiac causes group for prediction of poor outcome were 0.856 (95% CI: 0.836–0.875) and 0.805 (95% CI: 0.757–0.853), respectively. The cut-off values for each variable using Youden’s index were as follows: cardiac causes group, 138 μg/dL (sensitivity 0.76, specificity 0.82), and noncardiac causes group, 133 μg/dL (sensitivity 0.75, specificity 0.71) (Appendix A). Finally, the relationship between blood ammonia levels at hospital arrival and total prehospital time is shown for the cardiac causes and noncardiac causes groups (Appendix A). These results indicate that blood ammonia levels at hospital arrival may be a reliable predictor of neurological outcomes regardless of the cause of OHCA, whether cardiac or noncardiac. Furthermore, blood ammonia may have more precisely predicted neurological outcomes in the cardiac causes group compared with the noncardiac causes group.

## 4. Discussion

In this registry-based study of OHCA patients in Japan, we examined the hypothesis that blood ammonia levels at hospital arrival in conjugation with total prehospital time were superior to blood ammonia levels alone for prediction of favorable neurological outcomes at 30 days after OHCA. There may have been fewer patients with favorable neurological outcomes due to selection bias; however, blood ammonia levels alone were sufficient to identify those with favorable outcomes. The cut-off ammonia level for favorable outcomes in the overall study cohort was 138 μg/dL. This cut-off value varied depending on whether ROSC was achieved at the time of hospital arrival: it was 96.5 μg/dL for patients with ROSC at hospital arrival and 156 μg/dL for patients with sustained CA, respectively.

Given the assumption that blood ammonia levels would increase in proportion with ischemic duration, our results showed that the patients who had achieved ROSC had a lower cut-off ammonia level for favorable outcomes compared to patients with sustained CA. In a previous study, Shinozaki et al. found that blood ammonia levels at hospital admission was an independent predictor of favorable outcomes at six months after OHCA with a cut-off value of 170 μg/dL, which is a bit higher than the findings for our patients with sustained CA [16]. Similarly, other studies have revealed that blood ammonia levels below 100 μg/dL were associated with favorable outcomes at 30 days following OHCA [9,17], although these studies did not consider whether or not the patients had achieved ROSC at hospital arrival. Thus, our data suggest that blood ammonia levels should be interpreted based on additional findings, including the presence or absence of ROSC at hospital arrival.

A prior study showed that longer prehospital resuscitation time was associated with a lower chance of favorable neurological survival [18]. In contrast to this study, we observed that total prehospital time was unable to precisely predict patients with favorable outcomes. Although there is no plausible explanation for this discrepancy, the diagnostic performance of blood ammonia levels plus total prehospital time did not add any advantages in predicting favorable outcomes compared with blood ammonia levels alone. Prehospital time as a surrogate for ischemic duration was examined in only one study. Nagamine reported that blood ammonia levels were elevated in parallel with increased prehospital time; all patients in that study fully recovered without neurological sequelae unless ammonia levels exceeded 180 μg/dL [17].

Physical examinations, EEG, evoked potentials, imaging tests, and biomarkers are tools commonly used for neurological prognostication after CA. No single test among these is adequate for precise evaluation; multimodal prognostication is recommended to assess comatose patients [3]. Blood ammonia levels combined with lactate levels have previously been shown to improve prognostic performance. Additionally, it should be noted that blood ammonia levels did well at prognosticating neurological outcomes in children after CA, performing better than early EEG or computed tomography [19]. Measuring blood ammonia levels after OHCA can be useful as a multimodal method for determining neurological outcomes.

Ammonia, a component of the physiological buffer system that maintains pH homeostasis, is metabolized with the ornithine cycle, muscle, and astrocytes [19]. Serum ammonia can increase due to numerous causes. Patients with liver dysfunction cannot metabolize ammonia, leading to its accumulation. In patients with normal liver function, hyperammonemia can be due to drug toxicities and infections can be due to urease-producing bacteria, urea cycle disorders, acidosis, hyperalimentation, renal tubular acidosis, and increased muscle activity [20]. Likewise, blood ammonia levels increase in patients with CA due to both excessive ammonia production and metabolism dysfunction. CA and consequent interruption of blood flow for ammonia metabolization in tissue cause metabolic acidosis due to aerobic respiration due to oxygen deprivation and accumulation of end products such as hydrogen ions, lactate, and carbon dioxide. Respiratory and metabolic acidosis induce the release of ammonia from red blood cells, and ammonia damages the brain [21]. Furthermore, mitochondrial dysfunction associated with acidosis results in deterioration of glycolysis metabolism (the Krebs cycle) and the ornithine cycle [12]. Decreased metabolism results in persistent hyperammonemia, leading to a vicious cycle; hyperammonemia further damages the brain.

Hypoxia with CA increases blood ammonia concentration and brain cell damage associated with metabolic degradation. Based on evidence that hypoxic brain injury increases blood neuron specific enolase (NSE) concentrations [5,6,8], international guidelines recommend NSE use as part of a multimodal prognosis [8,22]. S100B is another biomarker often used for neuroprognostication in CA patients. However, the blood tests for NSE and S-100B have the disadvantage of lack of feasibility, because these tests are not always available in ordinary emergency laboratories. Ammonia is more easily measured than NSE and S-100B for prognostication in emergency centers.

The strength of this study was that it was a nationwide registry-based investigation on a larger population. Previous studies have already shown that blood ammonia levels were a strong predictor of favorable neurological outcomes [9,10,16,17]. Our study provides a confirmation on a rather large population though with a limited number of patients with favorable neurological outcomes. Further analysis was performed focusing on total prehospital time and ROSC at hospital arrival, which provided some novel insights into clinical interpretation. The cut-off ammonia levels values differed for patients with or without ROSC. Interestingly, total prehospital time did not have any impact on blood ammonia levels in patients with ROSC at hospital arrival. These findings should be considered when interpreting the ammonia levels of OHCA patients.

This study has some limitations. First, the influence of pre-existing diseases such as cardiovascular disease or chronic liver disease that potentially affect outcomes was not considered [23]. In particular, patients with underlying chronic liver disease may have had higher blood ammonia levels compared with those without underlying chronic liver disease. However, this can be almost ignored, presumably given the minority of these patients with chronic liver disease [24]. Second, there may have been a selection bias because the only patients with recorded blood ammonia levels were included in this investigation. Third, the quality of CPR during prehospital management was not considered. Fourth, the time at which blood samples were collected may have differed among the facilities, although overall they were obtained soon after hospital arrival. Fifth, there was little detailed information regarding hospital treatment. Sixth, the prehospital time was used to study this question, instead of low-flow time (from collapse to ROSC), because it was impossible to determine exact low-flow time of unwitnessed CA patients. Low-flow time may be longer than prehospital time. Finally, our findings may not be generalized outside of this population due to relatively lower numbers of patients with favorable neurological outcomes, presumably due to the specific EMS system in Japan and the age distribution of the Japanese population.

## 5. Conclusions

The results from our study on data from a large OHCA registry showed that blood ammonia level measured at hospital arrival can predict neurological prognosis at 30 days after OHCA, and these cut-off values were different for patients with ROSC and those without ROSC at hospital arrival. Blood ammonia levels gradually increased with increasing total prehospital time. However, the blood ammonia levels of the patients who had achieved ROSC at arrival were almost constant regardless of total prehospital time. The overall cut-off ammonia level for favorable outcomes was 138 μg/dL, and the cutoff values for patients with ROSC at hospital arrival was 96.5 μg/dL and 156 μg/dL for patients with sustained CA.

## Figures and Tables

**Figure 1 jcm-11-02566-f001:**
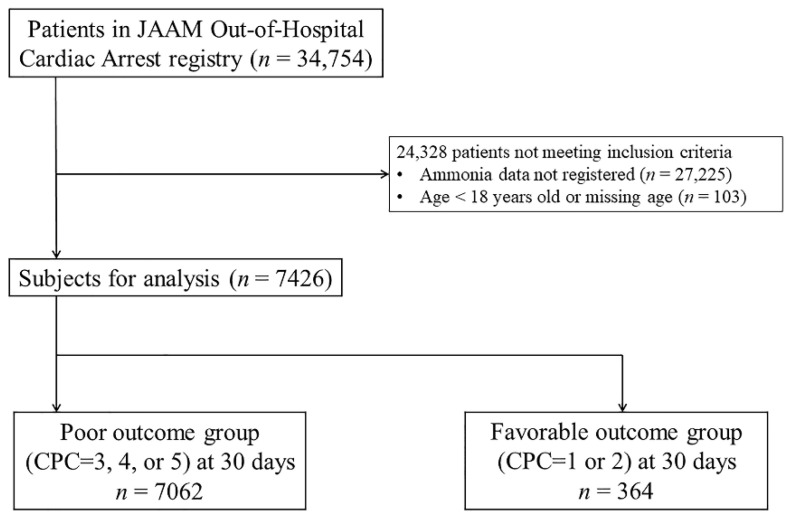
Flow diagram of patients analyzed. The favorable outcome group comprised patients with CPC scale scores of 1 or 2. The poor outcome group comprised patients with CPC scale sores of 3, 4, and 5. CPC: cerebral performance category, JAAM: Japanese Association for Acute Medicine.

**Figure 2 jcm-11-02566-f002:**
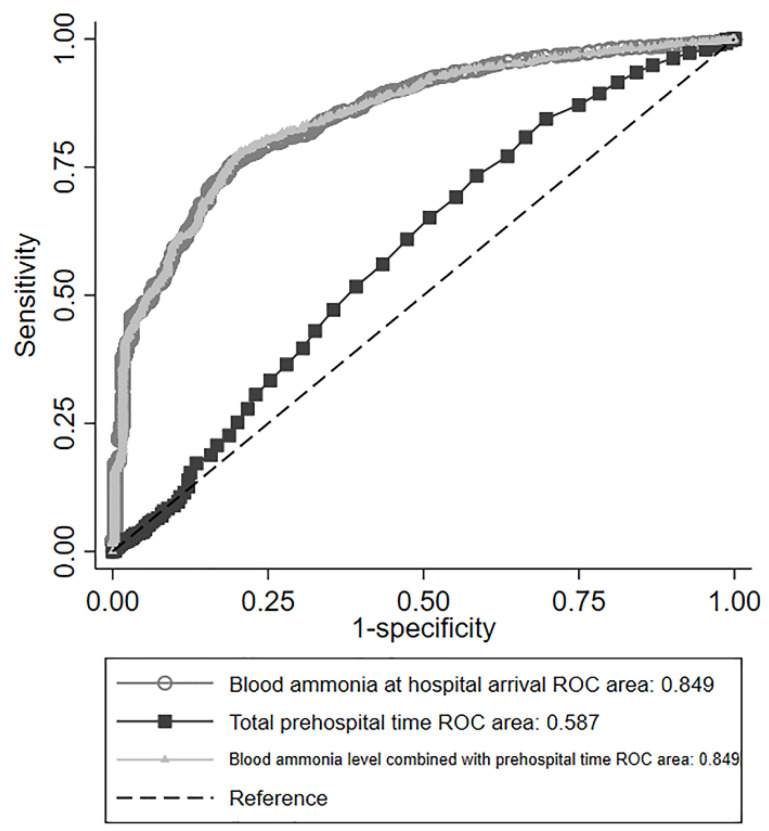
ROC curve for predicting poor outcomes with blood ammonia level (circle), total prehospital time (square), and blood ammonia level combined with total prehospital time (triangle). The areas under the ROC curve are shown in the box. The ROC curve of blood ammonia level at hospital arrival and of blood ammonia level combined with total prehospital time are almost identical. Abbreviations: receiver operating characteristic (ROC).

**Figure 3 jcm-11-02566-f003:**
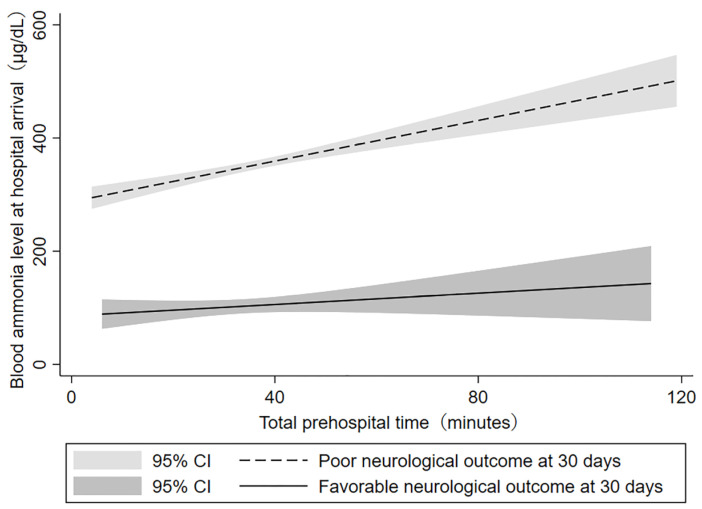
Relationship between blood ammonia level and total prehospital time. The blood ammonia level at hospital arrival increased with total prehospital time. The blood ammonia level in the favorable neurological outcomes group was lower than that in the poor neurological outcomes group. Abbreviations: confidence interval (CI).

**Figure 4 jcm-11-02566-f004:**
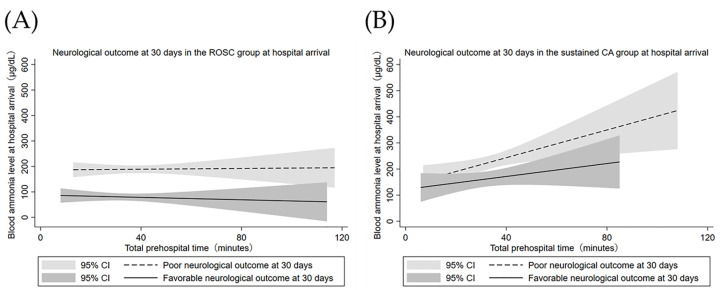
Relationship between blood ammonia level at hospital arrival and total prehospital time in the ROSC group (**A**) and the sustained CA group (**B**). In the ROSC patients, the blood ammonia levels of each outcome were almost constant regardless of total prehospital time. The blood ammonia level was lower in patients with favorable outcomes compared to those with poor outcomes. In the sustained CA patients, blood ammonia levels continued to increase in correlation with transportation time for both outcomes. Abbreviations: return of spontaneous circulation (ROSC); cardiopulmonary arrest (CA); confidence interval (CI).

**Table 1 jcm-11-02566-t001:** Patient characteristics and outcomes at 30 days.

	All Participants (*n* = 7426)	Favorable Outcomes (CPC 1–2) at 30 Days (*n* = 364)	Poor Outcomes (CPC 3–5) at 30 Days (*n* = 7062)	*p*-Value
Age (year), median (IQR)	76 (64–84)	63 (48–73)	76 (65–85)	<0.01
Male, *n* (%)	4538 (61%)	281 (77%)	4257 (60%)	<0.01
Total prehospital time * (min), median (IQR)	34 (28–41)	31 (25–39)	34 (28–42)	<0.01
Witnessed collapse, *n* (%)	3181 (43%)	259 (71%)	2922 (41%)	<0.01
Bystander CPR, *n* (%)	3202 (43%)	183 (50%)	3019 (43%)	<0.01
AED used by bystander, *n* (%)	149 (2.0%)	78 (21%)	71 (1.0%)	<0.01
Adrenaline used by EMS	2278 (31%)	35 (9.6%)	2243 (32%)	<0.01
Cardiac cause, *n* (%)	3719 (50%)	302 (83%)	3417 (48%)	<0.01
Initial rhythm at hospital arrival				
VF/pulseless VT, *n* (%)	392 (5.3%)	69 (19%)	323 (4.6%)	<0.01
PEA/asystole, *n* (%)	6124 (82%)	46 (13%)	6078 (86%)	<0.01
ROSC during transport, *n* (%)	910 (12%)	249 (68%)	661 (9.4%)	<0.01
Blood ammonia level (μg/dL), median (IQR)	253 (125–438)	70 (42–122)	265 (138–452)	<0.01
Potassium (mEq/L), median (IQR)	6.0 (4.6–7.9)	3.8 (3.4–4.2)	6.1 (4.8–8.1)	<0.01
Lactate (mg/dL), median (IQR)	118 (74–156)	64 (17–117)	119 (75–156)	<0.01
TTM, *n* (%)	634 (8.5%)	203 (56%)	431 (6.1%)	<0.01
Patients admitted to the hospital, *n* (%)	2457 (33%)	N/A	N/A	
Survival at 30 days, *n* (%)	738 (10%)	N/A	N/A	

Abbreviations: cardiopulmonary resuscitation (CPR); interquartile range (IQR); automated external defibrillator (AED); emergency medical services (EMS); ventricular fibrillation (VF); ventricular tachycardia (VT); pulseless electrical activity (PEA); return of spontaneous circulation (ROSC); targeted temperature management (TTM). * Total prehospital time was defined as the time span from awareness to hospital arrival.

**Table 2 jcm-11-02566-t002:** Univariable and multivariable logistic regression analysis of blood ammonia level and total prehospital time for favorable neurological outcomes.

	Crude OR (95% CI)	Adjusted OR (95% CI)
Blood ammonia level at hospital arrival	0.988 (0.987–0.990)	0.995 (0.994–0.998)
Total prehospital time	0.996 (0.996–0.997)	0.993 (0.981–1.006)

Abbreviations: odds ratio (OR); confidence interval (CI). Adjustments for multivariable logistic regression: age, sex, witnessed collapse, bystander cardiopulmonary resuscitation, automated external defibrillator use on scene, adrenaline use on scene, electrocardiogram on arrival, potassium and lactate levels at hospital arrival, and cardiogenic cause.

**Table 3 jcm-11-02566-t003:** Patient characteristics in the ROSC group and the sustained CA group at the time of hospital arrival.

	All Participants (*n* = 7426)	ROSC at Time of Hospital Arrival (*n* = 910)	Sustained CA at Hospital Arrival (*n* = 6516)	*p*-Value
Age (year), median (IQR)	76 (64–84)	74 (62–83)	76 (64–84)	0.012
Male, *n* (%)	4538 (61%)	576 (63%)	3962 (61%)	0.15
Total prehospital time * (min), median (IQR)	34 (28–41)	34 (29–41)	34 (28–41)	0.08
Witnessed collapse, *n* (%)	3181 (43%)	526 (58%)	2655 (41%)	<0.01
Bystander CPR, *n* (%)	3202 (43%)	411 (45%)	2791 (43%)	<0.01
AED used by bystander, *n* (%)	149 (2.0%)	76 (8.4%)	73 (1.1%)	<0.01
Adrenalin used by EMS	2278 (31%)	380 (42%)	1898 (29%)	<0.01
Cardiac cause, *n* (%)	3719 (50%)	426 (47%)	3293 (51%)	0.035
Initial rhythm at hospital arrival				
VF/pulseless VT, *n* (%)	392 (5.3%)	N/A	392 (6 %)	N/A
PEA/Asystole, *n* (%)	6124 (82%)	N/A	6124 (94%)	N/A
ROSC during transport, *n* (%)	910 (12%)	910 (100%)	N/A	N/A
Blood ammonia level (μg/dL), median (IQR)	253 (125–438)	103 (54–206)	278 (148–472)	<0.01
Potassium (mEq/L), median (IQR)	6.0 (4.6–7.9)	3.9 (3.5–4.8)	6.3 (4.9–8.2)	<0.01
Lactate (mg/dL), median (IQR)	118 (74–156)	81 (38–110)	119 (75–156)	<0.01
TTM, *n* (%)	634 (8.5%)	246 (27%)	388 (6.0%)	<0.01
Patients admitted to the hospital, *n* (%)	2457 (33%)	788 (87%)	1669 (26%)	<0.01
Survival at 30 days, *n* (%)	738 (10%)	419 (46%)	319 (4.9%)	<0.01
Favorable neurological outcome at 30 days, *n* (%)	364 (4.9%)	249 (27%)	115 (1.8%)	<0.01

Abbreviations: return of spontaneous circulation (ROSC); cardiac arrest (CA); interquartile range (IQR); cardiopulmonary resuscitation (CPR); automated external defibrillator (AED); emergency medical services (EMS); ventricular fibrillation (VF); ventricular tachycardia (VT); pulseless electrical activity (PEA); targeted temperature management (TTM). * Total prehospital time was defined as the time span from awareness to hospital arrival.

## Data Availability

The data in this study was from the Japanese Association for Acute Medicine (JAAM) OHCA Registry database. JAAM approves the sharing of data with interested researchers.

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
