# Peer review of "Can Blood Ammonia Level, Prehospital Time, and Return of Spontaneous Circulation Predict Neurological Outcomes of Out-of-Hospital Cardiac Arrest Patients? A Nationwide, Retrospective Cohort Study"

_jcm, 2022, doi:10.3390/jcm11092566_

Round 1

Reviewer 1 Report

The authors retrospectively investigated the effect of blood ammonia levels on the neurological outcome 30 days after OHCA in a large nationwide multicentric cohort study.

The paper is well written, and the sample size is large. The Cerebral performance category (CPC) scale score at 30 days evaluated neurological outcomes after OHCA. Prehospital data were selected in detail.

Blood ammonia levels were collected at hospital arrival. The authors showed that blood ammonia levels at hospital arrival could be a predictor of neurological prognosis at 30 days after OHCA with different cut-off values for the patients with or without ROSC at the time of hospital arrival and that blood ammonia level gradually increased with increasing total prehospital time which makes the data appear valid.

However, the reviewer suggests showing the p-values and the significance level in the patients’ characteristics and the regression analysis for a better comparison of groups. Furthermore, baseline characteristics might be extended by pre-existing diseases if available. Especially a pre-existing liver disease with chronic high ammonium levels would be of interest.

On top, the reviewer is interested if the results are the same for patients with cardiac cause and non-cardiac causes of OHCA.

Author Response

Reviewer 1

  • The reviewer suggests showing the p-values and the significance level in the patients’ characteristics and the regression analysis for a better comparison of groups. Furthermore, baseline characteristics might be extended by pre-existing diseases if available. Especially a pre-existing liver disease with chronic high ammonium levels would be of interest.

Authors’ comments
The authors agree with the reviewer’s comment. We added p-values for the patients’ characteristics and the regression analysis to Tables 1 and 3 for a better comparison of groups. Information on pre-existing diseases were not available in this registry. We are aware that the lack of data on pre-existing diseases is a considerable limitation of this study, which is stated in the revised manuscript.

“The registry does not include any information about comorbidities such as hypertension, diabetes, cardiovascular disease, chronic kidney disease, and chronic liver disease.” (lines 80-82)

“First, the influence of preexisting diseases such as cardiovascular disease or chronic liver disease that potentially affect outcomes was not considered [Hirlekar G, et al. Resuscitation. 2018:118-123.]. “In particular, the patients with underlying chronic liver disease may have possibly had higher blood ammonia levels compared with those without underlying chronic liver disease. However, this can be almost ignored, presumably given the minority of these patients with chronic liver disease [Roedl K, et al. Ann Intensive Care. 2017;7(1):103.] (lines 333-336)

  • On top, the reviewer is interested if the results are the same for patients with cardiac cause and non-cardiac causes of OHCA.

    Authors’ comments
    The authors appreciate these helpful comments. As requested, we conducted additional analyses and obtained several interesting data showing that blood ammonia levels at hospital arrival could predict neurological outcomes regardless of whether the cause of OHCA was cardiac or non-cardiac. (Tables S1, S2 and Figures S1, S2)

“3.5. Cardiac Causes vs. Non-Cardiac Causes

Further analysis was performed to determine whether there were differences between patients with OHCA from cardiac vs. those with OHCA from non-cardiac causes. Patients were divided into two groups: the cardiac causes group and the non-cardiac causes group. Patients’ characteristics are presented in Table S1. Multivariable logistic regression analysis revealed that lower blood ammonia levels were associated with favorable neurological outcomes in both the cardiac causes group and the non-cardiac causes group (Table S2). The areas under the ROC curve for the cardiac causes group and the non-cardiac causes group for prediction of poor outcome were 0.856 (95% CI: 0.836 – 0.875) and 0.805 (95% CI: 0.757 – 0.853), respectively. The cut-off values for each variable using Youden’s index were as follows: cardiac causes group, 138 μg/dL (sensitivity 0.76, specificity 0.82), and non-cardiac causes group, 133 μg/dL (sensitivity 0.75, specificity 0.71) (Figure S1). Finally, the relationship between blood ammonia levels at hospital arrival and total prehospital time is shown for the cardiac causes and non-cardiac causes groups (Figure S2). These results indicate that blood ammonia levels at hospital arrival may be a reliable predictor of neurological outcomes regardless of the cause of OHCA, whether cardiac or non-cardiac. Furthermore, blood ammonia may have more precisely predicted neurological outcomes in the cardiac causes group compared with the non-cardiac causes group.” (lines 239-257)

Reviewer 2 Report

My two main comments are:

  • There exists alreadyt knowlwdge on this topic.
  • ROC curves are a valuable tool to analyze test accuracy , however, if a certain parameter should be used to predict outcome after cardiac arrest, the analysis should focus on a cut-off value resulting in either in a high positive predictive value or a high negative predictive value. Given the very high prevalence of poor outcome in this patient population, probably a cut-off value resulting in a high positive predictive value for poor outcome should be identified.

Minor comments:

  • why are the data limited to 2017? What about data from 2017 to 2021?
  • Multivariate analysis should include all variables with difference in univariate analysis
  • The authors wanted to examine the impact of blood ammonia levels on patients’ outcome based on the assumption that sustained CA could lead to hyperammonemia. For this aim of the study, time from cardiac arrest to ROSC would be the correct parameter, not whether ROSC was achieved at admission. If patient is close to the hospital transported with chest compressions, achieving ROSC shortly after arrival, total CA time might be shorter as patients with long transportation time, achieving ROSC shortly before arrival

Author Response

We have described our response.

Please see the attachment and answers below.

Reviewer 2

  • ROC curves are a valuable tool to analyze test accuracy, however, if a certain parameter should be used to predict outcome after cardiac arrest, the analysis should focus on a cut-off value resulting in either in a high positive predictive value or a high negative predictive value. Given the very high prevalence of poor outcome in this patient population, probably a cut-off value resulting in a high positive predictive value for poor outcome should be identified.

Authors’ comments:
We agree with Reviewer 2 that PPV and NPV should be presented. As recommended, these two values were added for total prehospital time and ammonia levels. (lines 179-181)

  • Why are the data limited to 2017? What about data from 2017 to 2021?

Authors’ comments:
Thank you for bringing this to our attention. Unfortunately, the 2014 to 2017 data available to us was limited. The only data available for this period was distributed for our research purposes.

  • Multivariate analysis should include all variables with difference in univariate analysis

Authors’ comments:
Thank you for this remark. The Editor also suggested including several important variables. We have reanalyzed and modified Table 2 accordingly.

  • The authors wanted to examine the impact of blood ammonia levels on patients’ outcome based on the assumption that sustained CA could lead to hyperammonemia. For this aim of the study, time from cardiac arrest to ROSC would be the correct parameter, not whether ROSC was achieved at admission. If patient is close to the hospital transported with chest compressions, achieving ROSC shortly after arrival, total CA time might be shorter as patients with long transportation time, achieving ROSC shortly before arrival

Authors’ comments:
We understand the reviewer’s viewpoint. Technically, the time interval from collapse to ROSC might be preferable to test our hypothesis, but we focused on whether the patients had achieved ROSC at the time of hospital arrival as an alternative parameter because of the following two reasons. First, more than half of the patients in this study had unwitnessed collapse. Second, blood samples were uniformly taken at the time of hospital arrival. Given the data availability and explicitness, we believed that the parameter of whether patients had achieved ROSC would not be ideal and instead selected the best alternative parameter to examine how “low flow time” affected blood ammonia levels and ultimately, patients’ outcomes.

Reviewer 3 Report

Thank you very much for the opportunity of reviewing this paper indicating how high level of ammonia could discriminate poor neurologic outcome.

I have some comments:

1) it is not clear how you emergency system work. I guess you use to transport every patient to the hospital with ongoing resuscitation. Please provide us with  an accurate description of what you do in the field.

2) the rate of patients with a good neurologic outcome is rather low. Please spend some words about it

3) table 1 and three need an extra column with p values of the comparison between the two groups

4) you stated, and you are right, that other papers have reached similar results. Why are your results innovative. What did your paper add to modern knowledge on this topic? Please stress it more clearly in the discussion.

5) ROC curve graph should have a squared shape and not rectangular.

Author Response

We have described our response.

Please see the attachment and answers below.

Reviewer 3

  • It is not clear how your emergency system work. I guess you use to transport every patient to the hospital with ongoing resuscitation. Please provide us with an accurate description of what you do in the field.

Authors’ comments:
Thank you for asking for more details about the emergency system in Japan. You are correct. We transport all patients except those who are “obviously” deceased to the hospital with ongoing resuscitation. We have added detailed information on the emergency medical services system in Japan to the Materials and Methods section (lines 88-132).

“2.3. The Emergency Medical Services System in Japan

The EMS system in Japan has been described in detail elsewhere [15]. Briefly, all emergency calls (via 119 in Japan) are handled by local operations centers that dispatch the nearest ambulance to the scene. Each vehicle is staffed by three or four EMS personnel, at least one of whom is highly trained and known as an emergency life-saving technician (ELST). ELSTs can perform advanced airway management including supraglottic airway placement. In addition, specially trained ELSTs are allowed to perform endotracheal intubation and adrenaline administration under real-time medical direction by physicians. EMS personnel are obligated to resuscitate and transport OHCA patients to the hospital unless obvious signs of death are present.” (lines 88-97)

  • The rate of patients with a good neurologic outcome is rather low. Please spend some words about it

    Authors’ comments:
    There are several reasons why the number of patients with favorable neurological outcomes was low in this study. Two main reasons might explain this phenomenon: Japan’s EMS system and the age distribution of the Japanese population. As we mentioned above, EMS personnel must attempt resuscitation and transport patients to the hospital unless they have “obvious” signs of death. It is well-known that Japan is on the forefront of a super-aging society. Nonetheless, this fact potentially compromises the generalizability of this study. We have added the following sentence to the limitations discussion.

“Finally, our findings may not be generalized outside of this population due to relatively lower numbers of patients with favorable neurological outcomes, presumably due to the specific EMS system in Japan and the age distribution of the Japanese population.” (lines 341-344)

  • Table 1 and three need an extra column with p values of the comparison between the two groups

    Authors’ comments:
    We have added the p-values to Tables 1 and 3 as requested by the reviewer.

  • You stated, and you are right, that other papers have reached similar results. Why are your results innovative? What did your paper add to modern knowledge on this topic? Please stress it more clearly in the discussion.

    Authors’ comments: We have emphasized and amended the strong points of this paper as follows.

“A strength of this study was that it was based on a nationwide registry with a larger population. In line with previous studies, we found that blood ammonia levels were a strong predictor of favorable neurological outcomes. Further analysis was performed focusing on total prehospital time and whether ROSC was achieved at the time of hospital arrival, which provided some novel insights into clinical interpretation. Cut-off ammonia level values differ for patients with or without ROSC. Interestingly, total prehospital time did not have any impact on blood ammonia levels in patients who achieved ROSC at hospital arrival. These findings should be considered when interpreting the ammonia levels of OHCA patients.” (lines 324-331)

  • ROC curve graph should have a squared shape and not rectangular.

    Authors’ comments:
    As requested, we have made a correction on the ROC curve graph (Figure 2).

Round 2

Reviewer 2 Report

My main comments remain the same:

  • this study does not provide sufficient new knowledge to the readers of JCM.
  • For this research question, the Youdens index is not helpful. Given the very high prevalence of poor outcome in this patient population, probably a cut-off value resulting in a high positive predictive value for poor outcome should be identified.
  • Additionally, impact of blood ammonia levels on patients’ outcome based on the assumption that sustained CA could lead to hyperammonemia: the pre-hospital time is totally different from the low-flow time and cannot be used to study the research question

Author Response

I would appreciate it if you could see the attachment.

Reviewer 3 Report

Thank you very much for the effort in improving the quality of your paper. You did a good job and now the manuscript is clearer.

My only concern is that other studies have already pointed out the prognostic role of ammonia so it's not that innovative and original but it's provide  a confirmation on a rather large population though with a limited number of patients with good neurologic outcome.

Author Response

(The authors gave the same response as above.)
